# Disability, pain, and wound-specific concerns self-reported by adults at risk of limb loss: A cross-sectional study using the World Health Organization Disability Assessment Schedule 2.0

Derek J. Roberts[1,2]*, Sudhir K. Nagpal[1], Alan J. Forster[2,3,4,5], Timothy Brandys[1], Christine Murphy[1], Alison Jennings[2], Shira A. Strauss[1], Evgeniya Vishnyakova[1], Julie Lawson[2], Daniel I. McIsaac[2,3,5,6]

1 Division of Vascular and Endovascular Surgery, Department of Surgery, University of Ottawa, Ottawa, Ontario, Canada, 2 Clinical Epidemiology Program, Ottawa Hospital Research Institute, Ottawa, Ontario, Canada, 3 ICES, Toronto, Ontario, Canada, 4 Department of Medicine, University of Ottawa and The Ottawa Hospital, Ottawa, Ontario, Canada, 5 School of Epidemiology & Public Health, Faculty of Medicine, University of Ottawa, Ottawa, Ontario, Canada, 6 Departments of Anesthesiology and Pain Medicine, University of Ottawa and The Ottawa Hospital, Ottawa, Ontario, Canada

* Derek.Roberts01@gmail.com

## Abstract

### Introduction

There has been limited study of patient-reported outcomes (PROs) in patients at risk of limb loss. Our primary objective was to estimate the prevalence of disability in this patient population using the World Health Organization Disability Assessment Schedule 2.0 (WHODAS 2.0).

### Materials and methods

We recruited patients referred to a limb-preservation clinic. Patients self-reported their disability status using the 12-domain WHODAS 2.0. Severity of disability in each domain was scored from 1 = none to 5 = extreme and the total normalized to a 100-point scale (total score ≥25 = clinically significant disability). We also asked patients about wound-specific concerns and wound-related discomfort or distress.

### Results

We included 162 patients. Reasons for clinic referral included arterial-insufficient (37.4%), postoperative (25.9%), and mixed etiology (10.8%) wounds. The mean WHODAS 2.0 disability score was 35.0 (standard deviation = 16.0). One-hundred-and-nineteen (73.5%) patients had clinically significant disability. Patients reported they had the greatest difficulty walking a long distance (mean score = 4.2), standing for long periods of time (mean score = 3.6), taking care of household responsibilities (mean score = 2.7), and dealing with the emotional impact of their health problems (mean score = 2.5). In the two-weeks prior to

**Data Availability Statement:** Data are available from the authors for researchers who meet the

criteria for access to confidential data as approved by The Ottawa Health Science Network Research Ethics Board (which may be contacted at REBAdministration@toh.ca). Because the patient data included in the study are routinely collected data generated directly from clinical care, they are covered under Ontario's health privacy legislation, the Personal Health Information Protection Act (PHIPA). According to PHIPA, the authors are the custodians of these routinely collected data, and therefore they cannot make them openly available. Instead, the legislation requires the authors, as the data custodians, to consider each request on a case-by-case basis. Therefore, we will provide the data upon request to the principal author (Derek Roberts; Derek.Roberts01@gmail.com) because of the PHIPA.

**Funding:** DJR received funding from the Department of Surgery, University of Ottawa to support this work. The funders had no role in study design, data collection and analysis, decision to publish, or preparation of the manuscript.

**Competing interests:** The authors have declared that no competing interests exist.

presentation, 87 (52.7%) patients expressed concern over their wound(s) and 90 (55.6%) suffered a moderate amount or great deal of wound-related discomfort or distress. In adjusted ordinary least squares regression models, although WHODAS 2.0 disability scores varied with changes in wound volume (p = 0.03) and total revised photographic wound assessment tool scores (p<0.001), the largest decrease in disability severity was seen in patients with less wound-specific concerns and wound-related discomfort and distress.

## Discussion

The majority of people at risk of limb loss report suffering a substantial burden of disability, pain, and wound-specific concerns. Research is needed to further evaluate the WHODAS 2.0 in a multicenter fashion among these patients and determine whether care and interventions may improve their PROs.

## Introduction

Patient-reported outcomes (PROs) include measures of disability, health-related quality of life (HRQoL), and disease-specific concerns [1]. These measures inform patients, clinicians, and policy-makers about how patients feel or function in relation to their health condition without interpretation by healthcare providers and may identify targets for intervention [1, 2]. PROs are especially important for patients with chronic diseases [1, 2]. However, although patients at risk of limb loss because of hard-to-heal arterial-insufficient, venous, and diabetic foot ulcers (i.e., those wounds commonly seen by vascular surgeons and other wound care specialists) may suffer from disability or a reduced HRQoL [3–7], little data on PROs exist for these patients.

PRO instruments can be generic (i.e., relevant across disease states) or disease-specific (i.e., targeted to features of a given disease). In vascular surgery and medicine, there is presently no widely accepted instrument for assessing PROs in a generic or disease-specific manner across patients at risk of limb loss because of different types of hard-to-heal wounds [1, 8, 9]. However, systematic reviews of existing PRO instruments for patients with arterial-insufficient, venous, and diabetic foot ulcers have identified several domains that are commonly present across disease-specific PRO instruments and valued by patients with these wounds [3–7]. These include pain and other symptoms, activity limitations, and social and emotional impacts [3–7].

The World Health Organization (WHO) Disability Assessment Schedule 2.0 (WHODAS 2.0) is a broadly validated generic PRO instrument developed to measure disability across cultures and disease states [10, 11]. The WHO defines disability as "difficulties in any area of functioning as they relate to environmental and personal factors" [11, 12]. The WHODAS 2.0 has been shown to be a clinically acceptable, valid, reliable, and responsive instrument for measuring disability across a variety of acute and chronic surgical [11, 13–16] and medical [17–20] conditions. Many domains evaluated by the WHODAS 2.0 are relevant to patients at risk of limb loss (e.g., cognition, mobility, self-care, interpersonal relationships, work and household roles, and participation in society) [3–8, 10, 11]. Therefore, the WHODAS 2.0 could represent a relevant PRO instrument for these patients that permits a direct comparison with those with other medical conditions. However, to our knowledge, the WHODAS 2.0 has not been used to estimate disability among this patient population.

Given its broad validity, we hypothesized that the WHODAS 2.0 would be acceptable to patients at risk of limb loss when administered in a clinic setting and would allow for

quantification of the prevalence and severity of disability among these patients. We also hypothesized that the WHODAS 2.0 would provide unique patient-important information above that provided by characteristics of these patients and their wounds as well as a direct comparison of the degree of disability to those with other medical and surgical conditions. The primary objective of the study was to estimate the prevalence and severity of disability in patients at risk of limb loss using the WHODAS 2.0. Secondary objectives were to determine whether the WHODAS 2.0 was clinically acceptable to these patients and the extent to which patient and wound characteristics, wound-specific concerns, and wound-related discomfort or distress predicted their disability.

## Materials and methods

### Design, objectives, and ethics

We conducted a cross-sectional study. After review of our ethics submission, The Ottawa Health Science Network Research Ethics Board deemed that the study fell within the context of a quality initiative and waived the need for full ethics approval and patient consent. Reporting followed recommended guidelines [21, 22].

### Setting

The study was set at The Ottawa Hospital (TOH) Limb-Preservation Clinic. The clinic is affiliated with the University of Ottawa and run by a PhD (wound care)-trained specialty wound care nurse and the Division of Vascular and Endovascular Surgery in conjunction with colleagues from infectious diseases and orthopedic and plastic and reconstructive surgery. The clinic receives referrals from Division of Vascular and Endovascular Surgery faculty and house staff for patients with hard-to-heal wounds at risk of limb loss. It arranges in-person and virtual patient visits and follows patients in the outpatient and inpatient settings until their wounds are healed. Therapies provided in clinic include surgical and ultrasound debridement, negative-pressure wound therapy, total contact casting, skin grafting, and toe or ray foot amputations.

### Participants

There is no well-validated instrument for assessing risk of limb loss among patients with different wound types aside from arterial-insufficient and diabetic foot wounds. We therefore included consecutive adults (age >18-years) referred to TOH Limb-Preservation Clinic starting in June 1, 2018 thought to be at risk of limb loss by both the specialty wound care nurse and one of six vascular surgeons with extensive experience in limb-preservation. A vascular surgeon first evaluated all patients before they were seen in clinic. Our goal was to recruit a diverse cohort of patients with hard-to-heal wounds at risk of limb loss.

### Data collection, methods of measurement, and definitions

During the clinic visit, the specialty wound care nurse prospectively collected data on patient demographics, comorbidities, and prior revascularization therapies and assessed, measured, and photographed patients' wounds using how2trak® wound care software (Health Outcomes Worldwide, Toronto, ON, Canada). how2trak® is a wound care technology platform that includes longitudinal documentation of wounds, multidisciplinary assessments, individualized care plans, and two-way communication of wound status and treatment plan between referral centers and community providers. She also classified patients' wounds by assigning a number from 0 to 4 for each of the domains contained in the revised photographic wound

assessment tool (revPWAT) [23, 24]. revPWAT domains included wound size and depth, type and amount of necrotic and granulation tissue, wound edges, and periulcer skin viability [23, 24].

Upon clinic arrival, patients were also asked by a dedicated quality improvement coordinator to self-report their disability status using the 12-domain WHODAS 2.0 [10, 11]. The questionnaire was administered in the same clinic location under the same testing conditions and answers were entered into how2trak® by the quality improvement coordinator using a tablet. Each WHODAS 2.0 domain was scored on a 5-point Likert scale (none = 1; mild = 2; moderate = 3; severe = 4; and extreme = 5), as previously described [10, 11]. The score was then normalized to a 100-point scale (higher scores mean greater disability) [11]. A normalized score ≥25 defined clinically significant disability [25]. Disability was also further categorized into five ordinal categories based on normalized scores [none (0–4), mild (5–24), moderate (25–49), severe (50–95), and complete (96–100)] [25].

Based on published systematic reviews of PROs in vascular surgery [3–7], four of the six domains of the most widely validated peripheral artery disease-specific PRO (The Vascular Quality of Life Questionnaire-6 [26]) directly align with domains in the WHODAS 2.0. Therefore, we also asked patients to provide responses for the two disease-specific domains not covered by the WHODAS 2.0 (concern about their wound and wound-related discomfort or distress during the past two weeks). These responses were also scored on a 5-point Likert scale ranging from 1 = none of the time to 5 = all of the time for wound-specific concerns and 1 = none to 5 = a great deal for wound-related discomfort or distress.

We assessed the clinical acceptability of the WHODAS 2.0 by assessing completion rates and asking patients to answer four acceptability questions scored on 5-point Likert scales [27, 28]. These included how easy the questionnaire was to complete (which ranged from 1 = very easy to 5 = very hard), whether the questionnaire included important questions (which ranged from 1 = strongly disagree to 5 = strongly agree), whether their care may benefit from the information collected (which ranged from 1 = strongly disagree to 5 = strongly agree), and whether the questionnaire would be something they would be willing to do again (which ranged from 1 = extremely unwilling to 5 = extremely willing).

Finally, two investigators independently examined all physician and allied health service consults, laboratory results, and pharmacy information recorded in our hospital electronic medical record in duplicate during the last three years for each included patient. They then independently recorded data on comorbidities, medications, amputations, lower limb revascularization procedures, microbiology results, and hospitalizations and cross-referenced these with the data recorded in how2trak®. Discrepancies were resolved by consensus.

## Sample size

Our sample size was established to provide a precision of +/- 0.1 at the 5% significance level around a clinically-postulated proportion of prevalent disability assumed to be approximately 0.5. We estimated this to require 100 participants. However, as clinic referrals were high and data collection feasible, we collected data from study start until April 30, 2019 (the duration of time our quality improvement coordinator was available for the study).

## Statistical analyses

We summarized categorical data using counts (percentages), normally distributed continuous data using means and standard deviations (SD), and skewed data using medians with interquartile ranges (IQRs). Means and medians were compared using t-tests and Kruskal-Wallis tests, respectively. Covariate balance between those with and without disability were compared

using absolute standardized differences to avoid multiple hypothesis p-value testing; values >0.1 indicate substantive differences [29].

The clinical acceptability of the WHODAS 2.0 was analyzed by measuring the proportion of patients who completed the questionnaire during clinic visits, with the denominator representing all clinic patients approached by the quality improvement coordinator. We also created violin plots summarizing answers to the acceptability questions. Violin plots are modified box plots that add estimated kernel density plots to the summary statistics displayed by box plots. We estimated the proportion of patients who had clinically significant disability and calculated a 95% confidence interval (CI) around this estimate using Wilson's method [30]. A similar approach was used for disability severity categories and the proportion of patients with wound-related discomfort or distress and wound-specific concerns in the two-weeks prior to presentation.

To evaluate whether the WHODAS 2.0 added unique information for patients at risk of limb loss, we examined: 1) whether the WHODAS 2.0 score correlated with other patient-reported wound criteria (i.e., wound-specific concerns and wound-related discomfort or distress) and objective wound criteria (i.e., wound volumes and revPWAT scores) and 2) the extent to which WHODAS 2.0 scores were explained by baseline patient characteristics, patient-reported wound criteria, and objective wound criteria. If the WHODAS 2.0 added unique patient information, we would expect it to be moderately correlated and have its variance moderately explained by other characteristics, without being highly correlated or highly explained (which could suggest that each set of variables was measuring the very same construct). The correlations between the WHODAS 2.0 score and other wound criteria were evaluated using Pearson (for wound areas) or Spearman (for the other measures, which were ordinal in nature) coefficients.

The extent to which patient and wound characteristics, wound-specific concerns, and wound-related discomfort or distress explained the observed variance in WHODAS 2.0 scores was estimated by calculating $R^2$ values from ordinary least squares linear regression models. In these models, the WHODAS 2.0 score was the dependent variable and patient and wound characteristics, wound-specific concerns, and wound-related discomfort or distress were predictor variables. Three sets of models were created: 1) a multivariable model with patient characteristics as predictors; 2) a multivariable model with wound characteristics, wound-specific concerns, and wound-related discomfort or distress as predictors; and 3) a multivariable model with patient and wound characteristics, wound-specific concerns, and wound-related discomfort or distress as predictors.

Statistical analyses were conducted using Stata MP version 13.1 (Stata Corporation, College Station, Texas, U.S.A.).

## Results

### Patients and wounds

We included 162 consecutive patients at risk of limb loss due to hard-to-heal wounds (see Table 1 for characteristics of included patients and their wounds). The patients had a median age of 70.0 (IQR = 62.0–78.0) years and multiple comorbidities, including diabetes mellitus (57.4%) and coronary artery (42.6%) and chronic kidney (22.8%) disease. Further, 104 (64.6%) had undergone one ipsilateral lower limb revascularization procedure, 98 (60.5%) two or more ipsilateral revascularization procedures, and 10 (6.2%) a contralateral major amputation. Most had visited the Emergency Department (68%) or been hospitalized (78%) at least once within the last year, and 24% had been hospitalized more than once.

Reasons for clinic referral included arterial-insufficient (37.4%), postoperative (25.9%), mixed (e.g., arterial-insufficient and diabetic or chronic venous) (10.8%), venous (9.4%),

**Table 1. Baseline characteristics of the 162 patients at risk of limb loss.**

| Characteristic–No. (%) | Overall (n = 162) | Clinically Significant Disability (n = 119) | No Clinically Significant Disability (n = 43) | Absolute Standardized Difference[a] |
|---|---|---|---|---|
| *Personal characteristics* | | | | |
| Age, years–median (IQR) | 70.0 (62.0–78.0) | 70.0 (60.0–77.0) | 71.0 (66.0–78.0) | 0.18 |
| Male gender | 96 (59.3) | 70 (58.8) | 26 (60.5) | 0.03 |
| Rural residence | 67 (41.4) | 44 (37.0) | 23 (53.5) | 0.34 |
| Long-term care facility resident | 5 (3.1) | 5 (4.2) | 0 (0) | 0.30 |
| Current smoker | 38 (23.4) | 30 (25.2) | 8 (18.6) | 0.16 |
| Past smoker | 62 (38.3) | 70.0 (60.0–77.0) | 71.0 (66.0–78.0) | 0.18 |
| *Wound type (n = 139 with a documented etiology)* | | | | |
| Arterial | 52 (37.4) | 35/102 (34.3) | 17/37 (46.0) | 0.24 |
| Venous | 13 (9.4) | 7/102 (6.9) | 6/37 (16.2) | 0.30 |
| Diabetic | 10 (7.2) | 8/102 (7.8) | 2/37 (5.4) | 0.10 |
| Mixed | 15 (10.8) | 11/102 (10.8) | 4/37 (10.8) | 0.0009 |
| Postoperative | 36 (25.9) | 30/102 (29.4) | 6/37 (16.2) | 0.32 |
| Other | 13 (9.4) | 11/102 (10.8) | 2/37 (5.4) | 0.20 |
| *Wound location (n = 139 with a documented location)* | | | | |
| Groin | 3 (2.2) | 2/102 (2.0) | 1/37 (2.7) | 0.05 |
| Thigh | 9 (6.5) | 6/102 (5.9) | 3/37 (8.1) | 0.09 |
| Leg | 39 (28.1) | 30/102 (29.4) | 9/37 (24.3) | 0.11 |
| Foot or ankle | 85 (61.2) | 62/102 (60.8) | 23 (62.2) | 0.03 |
| Other | 3 (2.2) | 2/102 (2.0) | 1 (2.7) | 0.05 |
| *Wound size, cm–median (IQR)* | | | | |
| Width | 2.0 (1.0–4.0) | 1.8 (1.0–3.5) | 2.5 (1.2–5.0) | 0.38 |
| Length | 3.5 (1.7–6.0) | 3.2 (1.7–6.0) | 3.6 (1.9–7.5) | 0.24 |
| Depth | 0.40 (0.20–0.50) | 0.30 (0.20–0.50) | 0.40 (0.20–1.0) | 0.19 |
| Area | 7.5 (1.8–22.0) | 6.0 (2.0–18.0) | 8.6 (1.8–36.0) | 0.33 |
| Volume, $cm^3$ | 2.5 (0.54–11.1) | 2.2 (0.50–9.0) | 3.8 (0.75–16.8) | 0.26 |
| *Comorbidities recorded in the 3 years before clinic visit* | | | | |
| Acute coronary syndrome | 60 (37.0) | 46 (38.7) | 14 (32.6) | 0.13 |
| Cerebrovascular event (stroke or transient ischemic attack) | 22 (13.6) | 20 (16.8) | 2 (4.7) | 0.40 |
| Chronic kidney disease | 37 (22.8) | 27 (22.7) | 10 (23.3) | 0.01 |
| Chronic obstructive pulmonary disease | 21 (13.0) | 15 (12.6) | 6 (14.0) | 0.04 |
| Coronary artery disease | 69 (42.6) | 55 (46.2) | 14 (32.6) | 0.28 |
| Diabetes mellitus | 93 (57.4) | 74 (62.2) | 19 (44.2) | 0.37 |
| Dialysis | 14 (8.6) | 12 (10.1) | 2 (4.7) | 0.21 |
| Dyslipidemia | 99 (61.1) | 74 (62.2) | 25 (58.1) | 0.08 |
| Heart failure | 35 (21.6) | 30 (25.2) | 5 (11.6) | 0.36 |
| Hypertension | 126 (77.8) | 95 (79.8) | 31 (72.1) | 0.18 |
| *Prior lower limb amputation* | | | | |
| Contralateral above- or below-knee amputation | 10 (6.2) | 9 (7.6) | 1 (2.3) | 0.24 |
| Ipsilateral toe or ray amputation | 42 (25.9) | 30 (25.2) | 12 (27.9) | 0.14 |
| Ipsilateral transmetatarsal amputation | 11 (6.8) | 9 (7.6) | 2 (4.7) | 0.12 |
| *Ipsilateral lower limb revascularization procedures performed* | | | | |
| Iliac artery angioplasty and/or stenting | 23 (14.2) | 19 (16.0) | 4 (9.3) | 0.20 |

*(Continued)*

**Table 1.** (Continued)

| Characteristic–No. (%) | Overall (n = 162) | Clinically Significant Disability (n = 119) | No Clinically Significant Disability (n = 43) | Absolute Standardized Difference[a] |
|---|---|---|---|---|
| Femoral and/or popliteal artery angioplasty and/or stenting | 52 (32.1) | 40 (33.6) | 12 (27.9) | 0.12 |
| Tibial and/or peroneal artery angioplasty | 50 (30.9) | 36 (30.3) | 14 (23.3) | 0.05 |
| Aortofemoral or aortobifemoral bypass | 1 (0.6) | 1 (0.8) | 0 (0) | 0.13 |
| Axillofemoral or axillobifemoral bypass | 10 (6.2) | 7 (5.9) | 3 (7.0) | 0.04 |
| Iliofemoral or femoral endarterectomy | 27 (16.7) | 22 (18.5) | 5 (11.6) | 0.19 |
| Femoral-popliteal bypass | 12 (7.4) | 11 (9.2) | 1 (2.3) | 0.30 |
| Femoral-tibial or -peroneal bypass | 10 (6.2) | 7 (5.9) | 3 (7.0) | 0.04 |
| Other revascularization procedure | 21 (13.0) | 13 (10.9) | 8 (18.6) | 0.22 |

Where IQR indicates interquartile range.

[a]The absolute standardized difference allows comparison of the difference in prevalence of binary covariates, or the average of continuous covariates, between treatment groups without the influence of sample size. Values >0.1 indicate substantive differences [29].

diabetic (7.2%), and other (10.4%) wounds. Most wounds were located on the foot or ankle (61.2%). The wounds had a median volume of 2.5 (IQR = 0.54–11.1) cm$^3$, and nearly half (43.8%) had an associated wound infection. Wounds on the foot or ankle were most often on the toes (25.9%), heel (23.5%), or medial or lateral malleolus (10.6%). Median wound volumes did not vary significantly by wound type (p = 0.55) or location (p = 0.34). Patient revPWAT scores are summarized in S1 Table.

## Patient-reported disability, pain, and wound-specific concerns

The WHODAS 2.0 had excellent clinical acceptability: 100% of the 162 approached patients completed the questionnaire. Further, most patients thought the questionnaire was easy to complete, agreed that it included important questions and their care would benefit from information collected by the questionnaire, and were willing to do it again (Fig 1).

The mean overall WHODAS 2.0 disability score reported by the included patients was 35.0 (standard deviation = 16.0). Further, 119 (73.5%; 95% CI = 66.2–80%) suffered from clinically significant disability and most (54.3%; 95% CI = 46.6–61.8%) were moderately disabled (Fig 2). Patients' mean WHODAS 2.0 disability domain scores are summarized in Fig 3. Patients reported that during the past 30-days, they had severe difficulty walking a long distance (mean score = 4.2) and standing for long periods of time (mean score = 3.6). They also had mild-to-moderate difficulty taking care of their household responsibilities (mean score = 2.7), washing their whole body (mean score = 2.3), completing day-to-day work (mean score = 2.2), and joining in community activities (mean score = 2.0). Finally, they were moderately emotionally affected by their health problems during this time (mean score = 2.5).

In the two weeks prior to presentation, 87 (52.7%; 95% CI = 46.0–61.2%) patients expressed concern over their wound(s) at least a little of the time and 90 (55.6%; 95% CI = 47.9–63.0%) suffered a moderate amount or great deal of wound-related pain (Fig 4). Forty-four (27.2%; 95% CI = 20.9–34.5%) expressed concern over their wound(s) all of the time during this time period.

## WHODAS 2.0 disability scores in relation to patient characteristics and wound criteria

Correlations between WHODAS 2.0 disability scores and wound characteristics, wound-specific concerns, or wound-related discomfort and distress are reported in S2 Table. WHODAS

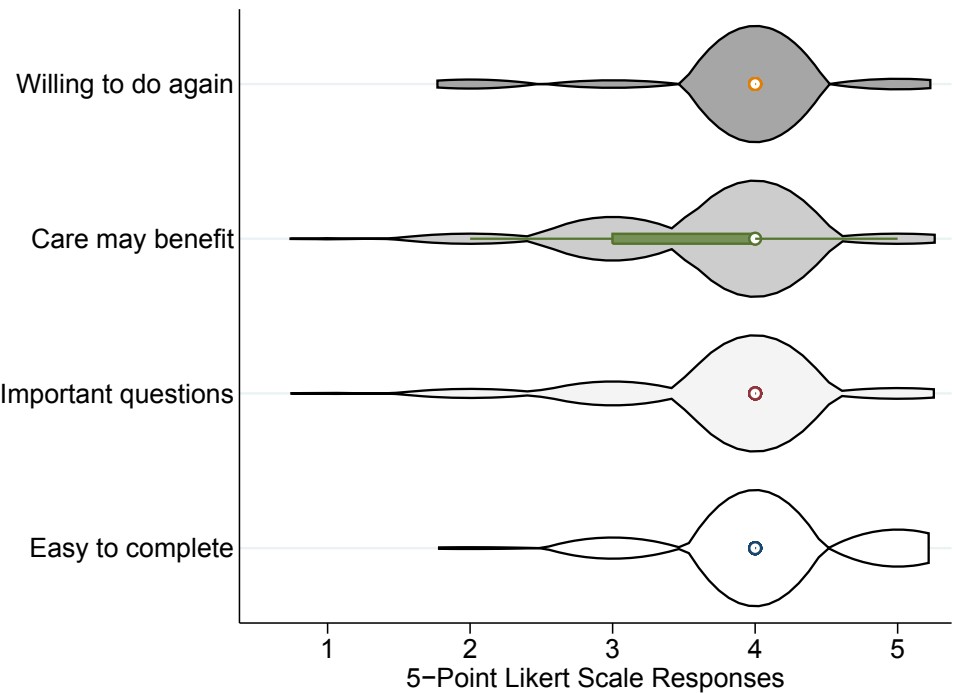

**Fig 1. Violin plots of answers to acceptability questions scored on 5-point Likert scales.** Violin plots are modified box plots that add estimated kernel density plots to the summary statistics displayed by box plots. The 5-point Likert scales ranged from 1 = very unwilling/strongly disagree/very hard to 5 = very willing/strongly agree/very easy.

2.0 disability scores were moderately correlated with total revPWAT scores (r = -0.42). They were also moderately correlated with the time patients' reported they spent concerned about their wounds (r = 0.26) and degree to which they reported suffering wound-related discomfort or distress (r = 0.23) during the past two-weeks. In contrast, WHODAS 2.0 disability scores were only weakly correlated with total wound volumes (r = -0.11).

Table 2 reports results of multivariable ordinary least squares linear regression analyses. In these analyses, models containing wound characteristics, wound-specific concerns, and wound-related discomfort or distress explained a larger proportion of variance in the model ($R^2$-value = 0.40) than did patient characteristics ($R^2$-value = 0.07). Further, adding patient characteristics into the model containing wound characteristics, wound-specific concerns, and wound-related discomfort or distress did not substantively change the proportion of variance explained by the model (i.e., the $R^2$-value changed from 0.40 to only 0.44 after patient characteristics were added). In the largest adjusted model containing patient characteristics, wound characteristics, wound-specific concerns, and wound-related discomfort or pain, WHODAS 2.0 disability scores were similar across patients with different ages, genders, comorbidities, and wound types and locations. They did vary somewhat with changes in wound volume (p = 0.03) and total revPWAT scores (p<0.001). However, the largest decrease in WHODAS 2.0 disability scores was seen in patients with less wound-specific concerns and wound-related discomfort or distress.

## Discussion

In this cross-sectional study of patients at risk of limb loss due to lower limb wounds, we found that almost three-out-of-four suffered from clinically significant disability. These

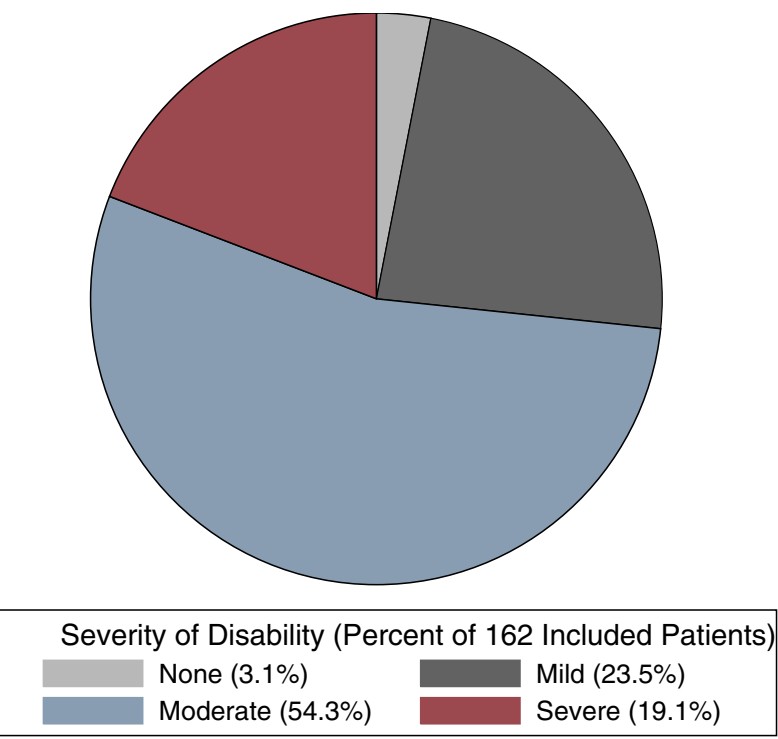

**Fig 2. Percent of 162 included patients with mild, moderate, and severe disability as defined by the World Health Organization Disability Assessment Schedule (WHODAS) 2.0.** Disability was also further categorized into five ordinal categories based on normalized scores [none (0–4), mild (5–24), moderate (25–49), severe (50–95), and complete (96–100)] [25].

patients had a number of different types of lower limb wounds, including arterial-insufficient, mixed, postoperative, chronic venous, and diabetic wounds. Further, over half of these patients expressed concerns over their wound(s) and suffered a moderate amount or great deal of wound-related discomfort or distress. Importantly, we found that the increasingly well-established clinical acceptability of the WHODAS 2.0 generalized to the older, comorbid patients routinely seen in a limb-preservation clinic. Finally, the WHODAS 2.0 had evidence of providing unique patient-important information as it was only moderately correlated with other patient-reported and objective wound criteria, and was not entirely explained by wound and patient characteristics.

We found that most of the included patients lived with moderate disability. Compared to the other populations that have been assessed using the WHODAS 2.0, patients at risk of limb loss reported disability scores exceeding those of medical and surgical patients, colorectal cancer patients with or without a stoma, multiple trauma and stroke victims, and patients with spinal cord injury [11, 13–20]. This is likely because most had severe difficulty performing even basic physical activity (e.g., standing or walking) and mild-to-moderate difficulty performing essential activities of daily living (e.g., taking care of their household responsibilities, washing their body, and completing day-to-day work). This highlights the substantial burden that hard-to-heal wounds place on patients, their families, and caregivers. These data also likely explain, at least in part, why most of the included patients reported being moderately emotionally affected by their health problems.

As this is the first study to examine use of the WHODAS 2.0 in patients at risk of limb loss, we are unable to directly compare our results to those of previously conducted studies.

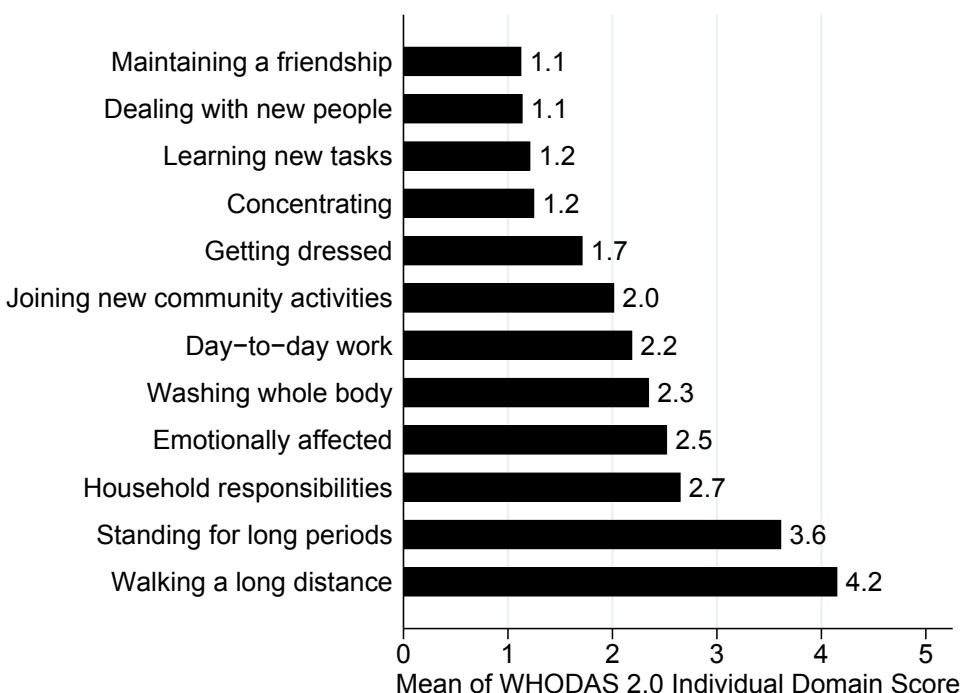

**Fig 3. Individual mean World Health Organization Disability Assessment Schedule (WHODAS) 2.0 disability domain scores self-reported by the 162 included patients.** Each domain of the WHODAS 2.0 was scored on a 5-point Likert scale (none = 1; mild = 2; moderate = 3; severe = 4; and extreme = 5), as previously described [10, 11].

However, a systematic review by Olsson *et al.* published in 2018 summarized results of studies examining the humanistic and economic burden of hard-to-heal arterial-insufficient, diabetic, and mixed-etiology lower extremity wounds as measured using nine different HRQoL instruments (most commonly the Short Form-36) and reported findings similar to ours [3]. Authors of this review included studies reported that published studies most commonly reported lower HRQoL scores in domains related to physical pathology and vitality and energy [3]. Further,

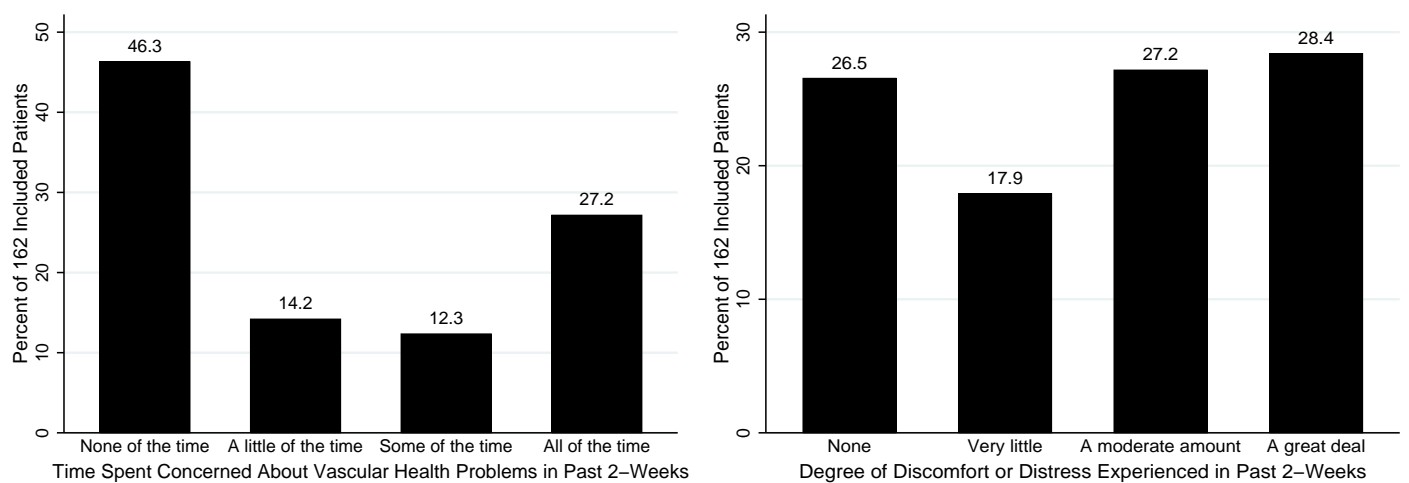

**Fig 4. Time spent concerned about their wound and degree of discomfort or distress experienced by the 162 included patients during the past two-weeks.**

**Table 2. Results of multivariable ordinary least squares linear regression models.**

| Predictor | Change in WHODAS 2.0 Score | | $R^2$-value for Model |
|---|---|---|---|
| | Adjusted Estimate (95% CI) | p-value | |
| *Model 1: Patient characteristics* | | | 0.074 |
| Age (per year increase) | -0.13 (-0.40 to 0.13) | 0.32 | |
| Male gender | -6.4 (-13.1 to 0.24) | 0.06 | |
| Comorbidities recorded in the 3-years before clinic visit | | | |
| Coronary artery disease | 0.93 (-5.7 to 7.6) | 0.78 | |
| Chronic kidney disease | -2.3 (-10.3 to 5.7) | 0.58 | |
| Chronic obstructive pulmonary disease | -11.3 (-24.6 to 1.9) | 0.09 | |
| Diabetes mellitus | 6.8 (-0.088 to 13.7) | 0.05 | |
| Hypertension | 4.2 (-3.8 to 12.1) | 0.31 | |
| *Model 2: Wound characteristics, wound-specific concerns, and wound-related discomfort or distress* | | | 0.40 |
| Wound type | | | |
| Arterial | 4.2 (-8.9 to 17.2) | 0.53 | |
| Venous | Reference | NA | |
| Diabetic | 12.6 (-4.9 to 30.1) | 0.16 | |
| Mixed | 7.1 (-6.8 to 21.0) | 0.31 | |
| Postoperative | 13.8 (-0.33 to 28.0) | 0.06 | |
| Other | 15.2 (0.23 to 30.3) | 0.05 | |
| Wound location | | | |
| Groin | -13.7 (-37.4 to 10.0) | 0.26 | |
| Thigh | Reference | NA | |
| Leg | 4.0 (-12.4 to 20.4) | 0.63 | |
| Foot or ankle | -0.40 (-14.9 to 14.1) | 0.96 | |
| Other | -24.7 (-52.2 to 2.7) | 0.08 | |
| Wound volume (per $cm^3$ increase) | -0.0086 (-0.017 to -0.00033) | 0.04 | |
| Total revPWAT score (per 1-point increase) | -1.2 (-1.6 to -0.78) | <0.001 | |
| Time spent concerned about their wound during the past 2-weeks | | | |
| None of the time | -8.6 (-16.3 to -0.93) | 0.03 | |
| A little of the time | 3.6 (-6.2 to 13.4) | 0.46 | |
| Some of the time | 4.3 (-6.7 to 15.2) | 0.44 | |
| All of the time | Reference | NA | |
| Degree of discomfort or distress experienced during the past 2-weeks | | | |
| None | -7.5 (-16.4 to 1.3) | 0.09 | |
| Very little | -12.1 (-21.6 to -2.6) | 0.01 | |
| A moderate amount | -3.7 (-12.7 to 5.3) | 0.42 | |
| A great deal | Reference | NA | |
| *Model 3: Patient and wound characteristics, pain, and disease-specific concerns* | | | 0.44 |
| Age (per year increase) | -0.14 (-0.45 to 0.16) | 0.35 | |
| Male gender | -5.2 (-12.2 to 1.9) | 0.15 | |
| Comorbidities recorded in the 3-years before clinic visit | | | |
| Coronary artery disease | 5.0 (-2.0 to 12.1) | 0.16 | |
| Chronic kidney disease | 0.19 (-8.1 to 8.4) | 0.96 | |
| Chronic obstructive pulmonary disease | 1.8 (-12.0 to 15.7) | 0.79 | |
| Diabetes mellitus | 2.8 (-5.0 to 10.5) | 0.48 | |
| Hypertension | 0.20 (-8.7 to 9.1) | 0.96 | |
| Wound type | | | |
| Arterial | 3.4 (-10.8 to 17.7) | 0.63 | |

(*Continued*)

**Table 2.** (Continued)

| Predictor | Change in WHODAS 2.0 Score | | R²-value for Model |
| --- | --- | --- | --- |
| | Adjusted Estimate (95% CI) | p-value | |
| Venous | Reference | NA | |
| Diabetic | 12.6 (-5.7 to 30.9) | 0.17 | |
| Mixed | 7.1 (-7.9 to 22.0) | 0.35 | |
| Postoperative | 13.4 (-1.5 to 28.3) | 0.08 | |
| Other | 14.6 (-1.3 to 30.5) | 0.07 | |
| Wound location | | | |
| Groin | -13.3 (-38.1 to 11.4) | 0.29 | |
| Thigh | Reference | NA | |
| Leg | 6.3 (-11.3 to 23.8) | 0.48 | |
| Foot or ankle | 1.8 (-14.4 to 18.1) | 0.82 | |
| Other | -19.7 (-48.8 to 9.4) | 0.18 | |
| Wound volume (per cm³ increase) | -0.0092 (-0.018 to -0.00067) | 0.03 | |
| Total revPWAT score (per 1-point increase) | -1.3 (-1.7 to -0.78) | <0.001 | |
| Time spent concerned about their wound during the past 2-weeks | | | |
| None of the time | -9.1 (-16.9 to -1.2) | 0.02 | |
| A little of the time | 1.7 (-8.5 to 12.0) | 0.74 | |
| Some of the time | 3.5 (-7.8 to 14.9) | 0.54 | |
| All of the time | Reference | NA | |
| Degree of discomfort or distress experienced during the past 2-weeks | | | |
| None | -8.4 (-17.5 to 0.74) | 0.07 | |
| Very little | -11.7 (-21.3 to -2.0) | 0.02 | |
| A moderate amount | -3.6 (-12.8 to 5.6) | 0.44 | |
| A great deal | Reference | NA | |

Where CI indicates confidence interval; revPWAT, revised photographic wound assessment tool; and WHODAS, World Health Organization Disability Assessment Schedule.

while studies reported mixed results regarding whether patients with hard-to-heal wounds had lower HRQoL scores in emotional and mental health domains, they did report that bodily pain and pain/discomfort scores were higher among those with hard-to-heal wounds than those who had undergone a major amputation [3].

Suggesting that the WHODAS 2.0 represents a useful PRO instrument that should be considered for use in clinical care, quality improvement, and research related to patients at risk of limb loss would require several conditions to be met. These include that it be acceptable to patients and provide unique information not available from other clinical measures. The scores provided should also validly reflect disease severity. In this study, we found evidence to support that the WHODAS 2.0 was clinically acceptable and that it had criterion validity among those at risk of limb loss. Interestingly, while WHODAS 2.0 disability scores were weakly correlated with patient wound volumes, they were moderately correlated with the appearance of the wound (i.e., revPWAT scores), wound-specific concerns, and wound-related discomfort or distress. Further, in multivariable regression analyses, wound characteristics, wound-specific concerns, and wound-related discomfort or distress explained a larger proportion of variance in the model than did patient characteristics, again supporting the relevance of the WHODAS 2.0 to patients at risk of limb loss. In a fully adjusted model containing patient and wound characteristics, wound-specific concerns, and wound-related discomfort or distress, patient's perceptions of their wounds (i.e., their concerns and discomfort or distress)

were more predictive of disability than information derived from the clinical history (i.e., age, gender, or associated comorbidities) or physical examination (i.e., wound volumes and revPWAT scores). This finding highlights the importance of PROs and their ability to quantify the impact of disease on patients' lived experiences.

Our findings need to be considered in the context of the study's limitations. First, we included patients referred to our limb-preservation clinic who were assessed to be at risk of limb loss by an experienced PhD (wound care)-trained specialist wound care nurse and vascular surgeon. Although the baseline characteristics of these patients appeared characteristic of patients at risk of limb loss, the Society for Vascular Surgery (SVS) Wound, Ischemia, foot Infection (SVS WIfI) classification system may have provided an additional objective assessment of the risk of limb loss among those with arterial-insufficient and diabetic wounds. However, the SVS WIfI was not designed for use in patients with other types of hard-to-heal wounds (e.g., chronic venous or postoperative wounds). Further, we lacked patient racial, economic, housing, and external social and financial support data, and some of those included in our study did not have arterial pressure measurements required to stratify patients into SVS WIfI stages at their first clinic assessment. Second, while our patients found the WHODAS 2.0 to be clinically acceptable, they did require assistance to input scores into a tablet. Future studies should therefore assess whether findings would be similar when collected via patient-facing data entry. Third, while it may be argued that a number of generic quality of life and disease-specific instruments already exist for assessing PROs in patients at risk of limb loss, the WHODAS 2.0 has been extensively validated, displays broad applicability across those at risk of limb loss, and when combined with measures of wound-specific concerns and discomfort/distress captures all of the domains covered by these other instruments. It also allows for direct comparison between other patient populations and studies of disability. Finally, while the WHODAS 2.0 is widely validated across disease states and demonstrated promising predictive validity in this study, our evaluation did not assess all aspects of validity. Future studies should therefore assess concurrent and convergent validity and perform longitudinal follow-up to determine reliability.

## Conclusions

In this cross-sectional study, we found that the majority of people at risk of limb loss suffered a substantial burden of disability. Most of them also expressed concern over their wounds and suffered a moderate amount or great deal of wound-related discomfort or distress. The WHODAS 2.0 displayed excellent clinical acceptability among the included patients as well as evidence of criterion validity. Therefore, it may provide additional important patient-centered information among patients at risk of limb loss. Future studies should further validate the instrument in a multicenter fashion. They should also determine how care and interventions (e.g., to enhance wound healing) provided over time to these patients may decrease their burden of disability, pain, and wound-specific concerns.

## Supporting information

**S1 Table. Included patient revised photographic wound assessment tool scores.**
(DOCX)

**S2 Table. Correlations/Associations between WHODAS 2.0 scores and wound characteristics and patient-reported pain and disease-specific concern scores.**
(DOCX)

## Author Contributions

**Conceptualization:** Derek J. Roberts, Sudhir K. Nagpal, Alan J. Forster, Timothy Brandys, Christine Murphy, Alison Jennings, Julie Lawson, Daniel I. McIsaac.

**Formal analysis:** Derek J. Roberts, Alan J. Forster, Christine Murphy, Alison Jennings, Shira A. Strauss, Daniel I. McIsaac.

**Funding acquisition:** Derek J. Roberts, Alan J. Forster.

**Investigation:** Derek J. Roberts, Sudhir K. Nagpal, Alan J. Forster, Timothy Brandys, Christine Murphy, Alison Jennings, Shira A. Strauss, Evgeniya Vishnyakova, Julie Lawson, Daniel I. McIsaac.

**Methodology:** Derek J. Roberts, Sudhir K. Nagpal, Alan J. Forster, Timothy Brandys, Christine Murphy, Alison Jennings, Shira A. Strauss, Evgeniya Vishnyakova, Julie Lawson, Daniel I. McIsaac.

**Project administration:** Derek J. Roberts, Alan J. Forster, Christine Murphy, Alison Jennings, Evgeniya Vishnyakova, Julie Lawson.

**Resources:** Derek J. Roberts, Sudhir K. Nagpal, Alan J. Forster, Timothy Brandys, Christine Murphy, Alison Jennings, Shira A. Strauss, Evgeniya Vishnyakova, Julie Lawson, Daniel I. McIsaac.

**Software:** Derek J. Roberts, Shira A. Strauss.

**Supervision:** Derek J. Roberts, Daniel I. McIsaac.

**Validation:** Shira A. Strauss, Evgeniya Vishnyakova, Julie Lawson.

**Writing – original draft:** Derek J. Roberts.

**Writing – review & editing:** Derek J. Roberts, Sudhir K. Nagpal, Alan J. Forster, Timothy Brandys, Christine Murphy, Alison Jennings, Shira A. Strauss, Evgeniya Vishnyakova, Julie Lawson, Daniel I. McIsaac.

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
