## [Decision Letter · Decision Letter 0]

27 Apr 2021

PONE-D-21-10113

Disability, pain, and wound-specific concerns self-reported by adults at risk of limb loss: A cross-sectional study

PLOS ONE

Dear Dr. Roberts,

Thank you for submitting your manuscript to PLOS ONE. After careful consideration, we feel that it has merit but does not fully meet PLOS ONE’s publication criteria as it currently stands. Therefore, we invite you to submit a revised version of the manuscript that addresses the points raised during the review process.

We look forward to receiving your revised manuscript.

Kind regards,

Kanhaiya Singh, Ph.D

Academic Editor

PLOS ONE

Journal Requirements:

2. Please provide additional details regarding participant consent. In the ethics statement in the Methods and online submission information, please ensure that you have specified (i) whether consent was informed and (ii) what type you obtained (for instance, written or verbal, and if verbal, how it was documented and witnessed). If your study included minors, state whether you obtained consent from parents or guardians. If the need for consent was waived by the ethics committee, please include this information.

3. In ethics statement in the manuscript and in the online submission form, please confirm that the IRB waived the need for full ethics approval.

Additional Editor Comments:

Reviewers have found this study interesting however they are recommending discussion of other confounding factors that may increase the risk of limb loss. Statistical analysis was done appropriately. Was power of this study calculated? Please also address the limitations of the present study.

Reviewers' comments:

Reviewer's Responses to Questions

**Comments to the Author**

1. Is the manuscript technically sound, and do the data support the conclusions?

Reviewer #1: Yes

Reviewer #2: Partly

2. Has the statistical analysis been performed appropriately and rigorously? 

Reviewer #1: I Don't Know

Reviewer #2: I Don't Know

3. Have the authors made all data underlying the findings in their manuscript fully available?

Reviewer #1: Yes

Reviewer #2: Yes

4. Is the manuscript presented in an intelligible fashion and written in standard English?

Reviewer #1: Yes

Reviewer #2: No

5. Review Comments to the Author

Reviewer #1: Ref: PONE-D-21-10113: needs Major revisions.

In the present article entitled “Disability, pain, and wound-specific concerns self-reported by adults at risk of limb loss: A cross-sectional study” Roberts, DJ et al., have estimated the burden of disability in patient-reported outcomes (PROs) in patients at risk of limb loss using the World Health Organization Disability Assessment Schedule 2.0 (WHODAS 2.0).

Although the manuscript contains a good amount of work, I have several concerns about the manuscript in its current form. Here are the specific comments.

1. Title: “WHODAS” maybe missing > may consider using WHODAS in second part of title instead of describing the methodology here.

2. Review of literature: Novelty concerns: a systematic review done recently by “Mundy et al.” clearly establishes an unmet need in the area, so this manuscript adds little in terms of providing a solution to the existing literature.

Mundy LR, Grier AJ, Weissler EH, Carty MJ, Pusic AL, Hollenbeck ST, Gage MJ. Patient-reported Outcome Instruments in Lower Extremity Trauma: A Systematic Review of the Literature. Plast Reconstr Surg Glob Open. 2019 May 3;7(5):e2218. doi: 10.1097/GOX.0000000000002218. PMID: 31333950; PMCID: PMC6571285.

3. Abstract: The result paragraph is not scientifically developed. Can it be explained in a more lucid manner? Example: well connected sentences.

4. Table 1: Personal characteristics or Demographics, line 219. Racial/ economic disparities could have been addressed.

5. Fig2: data representation can be better; may be a pie-diagram would be easier to interpret here as compared to bar.

6. While it is not a standard practice, NIH prefers scatter diagrams in the current grant guidelines as opposed to bar.

7. The inherent limitations of using a WHODAS scale such as bodily impairments and environmental factors have not been addressed. What has been done to address the high prevalence of co-morbid conditions in the study population or does WHODAS inherently address it?

8. WHODAS basically gives a qualitative scale 1-5; none, mild, moderate, severe, extreme. This is a subjective scale, is there be a more objective parameter to complement this data?

9. Will it be more prudent to follow up the patients since it is a well-established limb salvage clinic and look at the change in WHODAS (ΔWHODAS 2.0) with intervention as compared to a cross-sectional study?

Limb loss is a major concern in terms of its economic burden and also is a life-altering event. Hence I strongly encourage the authors to further work on this manuscript and I will be happy to provide further inputs.

Looking forward to your reply to these queries and hopefully this helps in improving the work.

Respectfully,

Reviewer

Reviewer #2: Ref: PONE-D-21-10113

In the present article entitled “Disability, pain, and wound-specific concerns self-reported by adults at risk of limb loss: A cross-sectional study” Roberts et al., have estimated the burden of disability in patient-reported outcomes (PROs) in patients at risk of limb loss using the World Health Organization Disability Assessment Schedule 2.0 (WHODAS 2.0). While the data collection and WHODAS2.0 application appear sound to me, here are specific comments.

1. Title can be made more precise and inclusion of WHODAS 2.0 to it can make it more apt.

2. Introduction: secondary objectives which were to determine the clinical acceptability of WHODAS 2.0, and to look for the extent to which various patient and wound characteristics can predict the degree of disability, can be included in the introduction.

3. Materials and methods: the patients recruited in this study were thought to be at the risk of limb loss. This can be elaborated further so as to mark if any objective criteria were followed to consider the probability of limb loss.

4. Data collection, methods of measurement, and definitions: Limb ischemia is one of the presentations of various systemic diseases like Diabetes, Atherosclerosis, Aortoarteritis, end stage renal disease and a few more. various other presentations and symptoms of these systemic illnesses can act as confounding factors while patients respond to the WHODAS2.0 questionnaire. So this aspect of confounding factors needs to be addressed further.

5. Discussion paragraph can be further delineated in an intelligible manner to improve the flow.

6. Authors have not discussed any limitations they faced or tackled during this study.

6. PLOS authors have the option to publish the peer review history of their article (what does this mean?). If published, this will include your full peer review and any attached files.

Reviewer #1: No

Reviewer #2: No

---

## [Author Response · Author response to Decision Letter 0]

19 May 2021

Derek J. Roberts, MD, PhD

Assistant Professor, Department of Surgery, University of Ottawa

The Ottawa Hospital, Civic Campus

Room A-280, 1053 Carling Avenue

Ottawa, Ontario, Canada

K1Y 4E9

Tel.: 613-798-5555, Ext. 16268

Fax: 613-761-5362

Derek.Roberts01@gmail.com

Kanhaiya Singh, PhD

Academic Editor, PLoS One

May 14, 2021

RE: Disability, pain, and wound-specific concerns self-reported by adults at risk of limb loss: A cross-sectional study using the World Health Organization Disability Assessment Schedule 2.0

Dear Dr. Singh:

Re:

Ms. No.: PONE-D-21-10113

Ms. Title: Disability, pain, and wound-specific concerns self-reported by adults at risk of limb loss: a cross-sectional study

We thank you and the PLoS One reviewers for the thoughtful reviews of our manuscript and the opportunity to resubmit a revised version for consideration of publication in PLoS One. We feel that our revised manuscript is an improved report, which we believe addresses each of your and the Reviewer’s comments point by point. Please find below an itemized list of detailed responses to each of your and the Reviewer’s comments, including a description of the changes made to the revised version of the manuscript (which are highlighted in yellow within the manuscript text). Within this itemized list, we first cited each comment verbatim before providing our response for ease of review.

**Comments from Dr. Singh:

1. Please ensure that your manuscript meets PLoS One’s style requirements, including those for file naming. The PLOS ONE style templates can be found at

 Thank you for the kind and thoughtful review of our manuscript. We have ensured that our manuscript meets PLoS One’s style requirements, including those for file naming, as outlined in the style templates outlined above.

2. Please provide additional details regarding participant consent. In the ethics statement in the Methods and online submission information, please ensure that you have specified (i) whether consent was informed and (ii) what type you obtained (for instance, written or verbal, and if verbal, how it was documented and witnessed). If your study included minors, state whether you obtained consent from parents or guardians. If the need for consent was waived by your ethics committee, please include this information. 

If you are reporting a retrospective study of medical records or achived samples, please ensure that you have discussed whether all data were fully anonymized before you accessed them and/or whether the IRB or ethics committee waived the requirement for informed consent. If patients provided informed written consent to have data from their medical records used in research, please include this information. 

Thank you. We submitted the protocol for the study to the Ottawa Health Science Network Research Ethics Board. Their review of the proposal indicated that the project fell within the context of a quality initiative. Consequently, they stated that as per the Tri-Council Policy Statement 2, Article 2.5, they waived the requirement for full ethics approval and patient consent. The study did not include minors. In order to better describe the above, we modified the “Design, objectives, and ethics” section of the Materials and methods. This now reads:

“Design, objectives, and ethics

 We conducted a cross-sectional study. After review of our ethics submission, The Ottawa Health Science Network Research Ethics Board deemed that the study fell within the context of a quality initiative and waived the need for full ethics approval and patient consent. Reporting followed recommended guidelines [1,2].” (paragraph 1, page 6)

3. In ethics statement in the manuscript and in the online submission form, please confirm that the IRB waived the need for full ethics approval.

Thank you. In the ethics statement in the manuscript and in the online submission form, we have now confirmed that the IRB waived the need for full ethics approval and patient consent (see above for a summary of the changes made to the Design, objectives, and ethics section of the Material and methods section of the manuscript).

 Thank you. The patients included in the study were those referred to a limb-preservation clinic who were thought to be at risk of limb loss by both a specialty wound care nurse and one of six vascular surgeons with extensive experience in limb-preservation. Because these patients have a visibly uncommon outcome, the Medical Director of The Ottawa Hospital limb-preservation clinic (Dr. S.K. Nagpal) and the Chief Innovation and Quality Officer at The Ottawa Hospital (Dr. A.J. Forster) felt that their patient information could not be sufficiently de-identified if it was presented on a non-summarized level. Further, because their data are derived from routinely collected data generated directly from clinical care, they are covered under Ontario’s health privacy legislation, the Personal Health Information Protection Act (PHIPA). According to PHIPA, as we are the custodians of these routinely collected data, we cannot make them openly available. Instead, the legislation requires us, as the data custodians, to consider each request on a case-by-case basis. Therefore, we had originally stated that we would provide the data upon request to the principal author (Derek Roberts; Derek.Roberts01@gmail.com) because of the PHIPA. We hope that this explains why we had indicated that the data would be provided upon request. We have also updated the above in the “Data Availability” section of the manuscript submission and provided e-mail addresses for contact.

 The PHIPHA Act states:

“Ontario’s health privacy legislation, the Personal Health Information Protection Act (PHIPA), establishes a set of rules regarding personal health information

Health information custodians who have custody or control of your personal health information are required to: designate or take on the role of a contact person to: 

• help the custodian to comply with their obligations under PHIPA,

• ensure that agents of the custodian are appropriately informed of their duties,

• respond to inquiries from the public about their information practices,

• respond to your requests for access and corrections to your information,

• receive complaints about alleged breaches of PHIPA”

5. Additional Editor Comments:

Reviewers have found this study interesting however they are recommending discussion of other confounding factors that may increase the risk of limb loss. Statistical analysis was done appropriately. Was power of this study calculated? Please also address the limitations of the present study.

Thank you. We agree with the Editor that many factors could influence an individual’s disability state, including specific health conditions (in our case, the presence of a lower extremity wound and/or their comorbidities), as well as social, economic or environmental factors. While in the current study we did not have access to socioeconomic or environmental factors (we assume the reviewer is likely referring to the built environment, like housing and support?), we do acknowledge this in the limitations section of the Discussion. Regarding comorbidities versus limb wounds, this was one of our secondary objectives, and as reported in Table 2, patient characteristics (age, sex, comorbidities) explained little of the variation in WHODAS disability scores (~7%), whereas objective and patient-reported wound characteristics explained a substantial amount (~40%). While this still leaves 60% to be explained, and we agree this could be partly explained by socioeconomic and environmental factors, we do hope that the reviewer agrees that we attempted to investigate these phenomena within the limitations of the data available to us. 

The power of the study was calculated a priori around the primary objective of estimating the prevalence of clinically significant disability in our target population. Our sample size was established to provide a precision of +/- 0.1 at the 5% significance level around a clinically-postulated proportion of prevalent disability assumed to be approximately 0.5. We estimated this to require 100 participants. However, as clinic referrals were high and data collection feasible, we collected data from study start until April 30, 2019 (the duration of time our quality improvement coordinator was available for the study). This is described on page 9 of the revised version of the manuscript. We have also included a limitations paragraph in the Discussion of the revised manuscript. This paragraph now reads:

“Our findings need to be considered in the context of the study’s limitations. First, we included patients referred to our limb-preservation clinic who were assessed to be at risk of limb loss by an experienced PhD (wound care)-trained specialist wound care nurse and vascular surgeon. Although the baseline characteristics of these patients appeared characteristic of patients at risk of limb loss, the Society for Vascular Surgery (SVS) Wound, Ischemia, foot Infection (SVS WIfI) classification system may have provided an additional objective assessment of the risk of limb loss among those with arterial-insufficient and diabetic wounds. However, the SVS WIfI was not designed for use in patients with other types of hard-to-heal wounds (e.g., chronic venous or postoperative wounds). Further, we lacked patient racial and economic data and some of those included in our study did not have arterial pressure measurements required to stratify patients into SVS WIfI stages at their first clinic assessment. Second, while our patients found the WHODAS 2.0 to be clinically acceptable, they did require assistance to input scores into a tablet. Future studies should therefore assess whether findings would be similar when collected via patient-facing data entry. Third, while it may be argued that a number of generic quality of life and disease-specific instruments already exist for assessing PROs in patients at risk of limb loss, the WHODAS 2.0 has been extensively validated, displays broad applicability across those at risk of limb loss, and when combined with measures of wound-specific concerns and discomfort/distress captures all of the domains covered by these other instruments. It also allows for direct comparison between other patient populations and studies of disability. Finally, while the WHODAS 2.0 is widely validated across disease states and demonstrated promising predictive validity in this study, our evaluation did not assess all aspects of validity. Future studies should therefore assess concurrent and convergent validity and perform longitudinal follow-up to determine reliability.” (paragraph 2, page 23-24)

***Reviewers’ comments:

Reviewer #1: Ref: PONE-D-21-10113: needs Major revisions.

In the present article entitled “Disability, pain, and wound-specific concerns self-reported by adults at risk of limb loss: a cross-sectional study” Roberts, DJ et al., have estimated the burden of disability in patient-reported outcomes (PROs) in patients at risk of limb loss using the World Health Organization Disability Assessment Schedule 2.0 (WHODAS 2.0).

Although the manuscript contains a good amount of work, I have several concerns about the manuscript in its current form. Here are the specific comments.

 We thank the Reviewer for their kind and thoughtful review of our manuscript. The comments have improved the study and are greatly appreciated.

1. Title: “WHODAS” maybe missing > may consider using WHODAS in second part of title instead of describing the methodology here.

We agree. We have therefore added the World Health Organization Disability Assessment Schedule to the second part of the title of the manuscript as suggested by the Reviewer. The revised title of the manuscript is:

“Disability, pain, and wound-specific concerns self-reported by adults at risk of limb loss: a cross-sectional study using the World Health Organization Disability Assessment Schedule 2.0”

2. Review of literature: Novelty concerns: a systematic review done recently by “Mundy et al.” clearly establishes an unmet need in the area, so this manuscript adds little in terms of providing a solution to the existing literature. Mundy LR, Grier AJ, Weissler EH, Carty MJ, Pusic AL, Hollenbeck ST, Gage MJ. Patient-reported Outcome Instruments in Lower Extremity Trauma: A Systematic Review of the Literature. Plast Reconstr Surg Glob Open. 2019 May 3;7(5):e2218. doi: 10.1097/GOX.0000000000002218. PMID: 31333950; PMCID: PMC6571285.

We thank the Reviewer for highlighting the above systematic review of patient-reported instruments in lower extremity trauma, and apologize if we do not understand the concerns regarding novelty. In the above systematic review, the authors sought to identify and evaluate patient-reported outcome (PRO) instruments developed specifically for lower extremity trauma that were applicable to both lower extremity reconstruction and amputation. These authors searched for English-language publications that described the development and/or validation of a PRO instrument assessing satisfaction and quality of life in lower extremity trauma patients. After reviewing 6,290 abstracts and 657 full-text citations, the authors were unable to identify any relevant studies. Therefore, they suggested that there was a need for a PRO instrument to better understand those with limb-threatening trauma.

We agree with the authors of this systematic review that there is no widely accepted instrument for assessing PROs in a generic or disease-specific manner across patients at risk of limb loss [3-5]. While we did not aim to evaluate those with lower extremity trauma (and no patients with lower extremity trauma were included in our study), our study did include those who are most commonly at risk of limb loss (i.e., those with arterial-insufficient, diabetic, and other types of hard-to-heal wounds) [6-10]. We hypothesized that the WHODAS 2.0 would be acceptable to patients at risk of limb loss when administered in a clinic setting and would allow for quantification of the prevalence and severity of disability among these patients. We also hypothesized that the WHODAS 2.0 would provide unique patient-important information above that provided by characteristics of these patients and their wounds as well as a direct comparison of the degree of disability to those with other medical and surgical conditions. 

To our knowledge, our study is novel as it is the first to study disability among patients at risk of limb loss using the WHODAS 2.0. This statement is supported by the findings of a recent systematic review of the economic and humanistic burden of hard-to-heal wounds by Ollson et al. in 2018 [6]. We found that the majority of people at risk of limb loss suffered a substantial burden of disability. Most of them also expressed concern over their wounds and suffered a moderate amount or great deal of wound-related discomfort or distress. The WHODAS 2.0 displayed excellent clinical acceptability among the included patients as well as evidence of criterion validity. Our study therefore has important implications for practice and future research. First, it suggests that the WHODAS 2.0 provides important patient-centered information among patients at risk of limb loss. Second, as the tool has been shown to be a clinically acceptable, valid, reliable, and responsive instrument for measuring disability across a variety of chronic surgical [11-15] and medical [16-19] conditions, our results suggest that measuring disability with this tool permits a direct comparison with those with other medical conditions. 

3. Abstract: The result paragraph is not scientifically developed. Can it be explained in a more lucid manner? Example: well connected sentences.

Thank you. We reviewed the results paragraph of the Abstract and ensured that it was scientifically developed and explained with well-connected sentences. The revised Abstract is listed below, which has been extensively edited in direct response to the above comment by the Reviewer. We feel that the Abstract now has well-connected sentences while still being concise and following PLoS One’s instructions for authors (the Abstract word limit for the Journal is 300 words). However, should the Reviewer or Editor wish that we modify any of the remaining sentences, we would be more than happy to do so. 

“Abstract

Introduction: There has been limited study of patient-reported outcomes (PROs) in patients at risk of limb loss. Our primary objective was to estimate the prevalence of disability in this patient population using the World Health Organization Disability Assessment Schedule 2.0 (WHODAS 2.0). 

Materials and methods: We recruited patients referred to a limb-preservation clinic. Patients self-reported their disability status using the 12-domain WHODAS 2.0. Severity of disability in each domain was scored from 1=none to 5=extreme and the total normalized to a 100-point scale (total score ≥25=clinically significant disability). We also asked patients about wound-specific concerns and wound-related discomfort or distress. 

Results: We included 162 patients. Reasons for clinic referral included arterial-insufficient (37.4%), postoperative (25.9%), and mixed etiology (10.8%) wounds. The mean WHODAS 2.0 disability score was 35.0 (standard deviation=16.0). One-hundred-and-nineteen (73.5%) patients had clinically significant disability. Patients reported they had the greatest difficulty walking a long distance (mean score=4.2), standing for long periods of time (mean score=3.6), taking care of household responsibilities (mean score=2.7), and dealing with the emotional impact of their health problems (mean score=2.5). In the two-weeks prior to presentation, 87 (52.7%) patients expressed concern over their wound(s) and 90 (55.6%) suffered a moderate amount or great deal of wound-related discomfort or distress. In adjusted ordinary least squares regression models, although WHODAS 2.0 disability scores varied with changes in wound volume (p=0.03) and total revised photographic wound assessment tool scores (p<0.001), the largest decrease in disability severity was seen in patients with less wound-specific concerns and wound-related discomfort and distress. 

Discussion: The majority of people at risk of limb loss report suffering a substantial burden of disability, pain, and wound-specific concerns. Research is needed to further evaluate the WHODAS 2.0 in a multicenter fashion among these patients and determine whether care and interventions may improve their PROs.”

4. Table 1: Personal characteristics or Demographics, line 219. Racial/economic disparities could have been addressed.

We unfortunately did not collect patient racial and economic data. These data are not routinely collected in how2trak or our hospital electronic medical record. However, as we agree with the Reviewer that these data may have added additional important information about the patients included in the study, we have added this as a limitation of the study in the limitations section of the manuscript. The limitations section of the manuscript now reads:

“Our findings need to be considered in the context of the study’s limitations. First, we included patients referred to our limb-preservation clinic who were assessed to be at risk of limb loss by an experienced PhD (wound care)-trained specialist wound care nurse and vascular surgeon. Although the baseline characteristics of these patients appeared characteristic of patients at risk of limb loss, the Society for Vascular Surgery (SVS) Wound, Ischemia, foot Infection (SVS WIfI) classification system may have provided an additional objective assessment of the risk of limb loss among those with arterial-insufficient and diabetic wounds. However, the SVS WIfI was not designed for use in patients with other types of hard-to-heal wounds (e.g., chronic venous or postoperative wounds). Further, we lacked patient racial, economic, housing, and external social and financial support data, and some of those included in our study did not have arterial pressure measurements required to stratify patients into SVS WIfI stages at their first clinic assessment. Second, while our patients found the WHODAS 2.0 to be clinically acceptable, they did require assistance to input scores into a tablet. Future studies should therefore assess whether findings would be similar when collected via patient-facing data entry. Third, while it may be argued that a number of generic quality of life and disease-specific instruments already exist for assessing PROs in patients at risk of limb loss, the WHODAS 2.0 has been extensively validated, displays broad applicability across those at risk of limb loss, and when combined with measures of wound-specific concerns and discomfort/distress captures all of the domains covered by these other instruments. It also allows for direct comparison between other patient populations and studies of disability. Finally, while the WHODAS 2.0 is widely validated across disease states and demonstrated promising predictive validity in this study, our evaluation did not assess all aspects of validity. Future studies should therefore assess concurrent and convergent validity and perform longitudinal follow-up to determine reliability.” (paragraph 2, page 23-24)

5. Fig2: data representation can be better; may be a pie-diagram wound be easier to interpret here as compared to bar.

We agree. In direct response to the above suggestion by the Reviewer, we therefore changed Fig. 2 from a bar diagram to a pie-diagram.

6. While it is not a standard practice, NIH prefers scatter diagrams in the current grant guidelines as opposed to bar.

Thank you for this comment. Fig. 3 could potentially be displayed as a scatter diagram while we do not believe that Fig. 4 could (the data are dichotomous so a scatter diagram would just show two dichotomous areas of bubbles representing the values “yes” and “no”). However, as there are 12 different variables displayed in the plot in Fig. 3, we would respectfully argue that the number of scatter bubbles would make the plot rather busy and difficult for readers to easily comprehend. Further, in our utilized statistical software (Stata), scatter plots can only be created as a twoway (y, x) plot that compares the relationship between two continuous or measured variables. We therefore elected to keep the bar diagram presented in the first version of the manuscript. However, if the Reviewer or Editor feel strongly about using a scatter instead of bar diagram for Fig. 3, we can submit this using an alternate statistical software program.

7. The inherent limitations of using a WHODAS scale such as bodily impairments and environmental factors have not been addressed. What has been done to address the high prevalence of co-morbid conditions in the study population or does WHODAS inherently address it?

Thank you for this comment. We hope that in responding we are properly interpreting the Reviewer’s comment. As a patient-reported outcome, the WHODAS allows quantification of the impact of a patient’s health state on their day to day function (or conversely, their degree of disability) using 12 domains that are highly relevant to patients and which are directly informed and linked to concepts in the International Classification of Functioning, Disability and Health (ICF). This means that the impact of health conditions can be quantified in a meaningful way without interpretation by a clinician or researcher. Therefore, we understand the greatest value in using the WHODAS 2.0 to be the ability to measure a patient’s self-reported disability status directly.

However, we agree with the reviewer that many factors could influence an individual’s disability state, including specific health conditions (in our case, the presence of a lower extremity wound and/or their comorbidities), as well as social, economic or environmental factors. While in the current study we did not have access to socioeconomic or environmental factors (we assume the reviewer is likely referring to the built environment, like housing and support?), we do acknowledge this in the limitations section of the Discussion. Regarding comorbidities versus limb wounds, this was one of our secondary objectives, and as reported in Table 2, patient characteristics (age, sex, comorbidities) explained little of the variation in WHODAS disability scores (~7%), whereas objective and patient-reported wound characteristics explained a substantial amount (~40%). While this still leaves 60% to be explained, and we agree this could be partly explained by socioeconomic and environmental factors, we do hope that the reviewer agrees that we attempted to investigate these phenomena within the limitations of the data available to us. 

8. WHODAS basically gives a qualitative scale 1-5; none, mild, moderate, severe, extreme. This is a subjective scale, is there be a more objective parameter to complement these data?

The WHODAS 2.0 collects measures of disability that are self-reported by the patient. The instrument uses 5-point scales to help to quantify the impact of each of the assessed domains on an individual’s disability. Furthermore, as a patient-reported outcome there is no need for this to be ‘filtered’ by the clinical team, meaning that these patient-reported outcome measures inform patients, clinicians, and policy-makers about how patients feel or function in relation to their health condition or conditions without interpretation by healthcare providers. 

While one could interpret this as subjective, we would also like to point to the psychometrics reported from the development of this instrument (which are provided by Ustun and colleagues [20]). These data demonstrate that the WHODAS 2.0 was found to have high internal consistency (Cronbach’s alpha, α: 0.86), a stable factor structure; high test-retest reliability (intraclass correlation coefficient: 0.98); good concurrent validity in patient classification when compared with other recognized disability measurement instruments; conformity to Rasch scaling properties across populations, and good responsiveness (i.e. sensitivity to change). In other words, it does appear that this approach to measuring disability in a quantitative but patient-reported manner is likely numerically valid.

Finally, in our study, these data appeared to provide unique patient-important information above standard clinical information as WHODAS 2.0 disability scores were only moderately correlated with objective wound criteria and were not entirely explained by wound and patient characteristics. Therefore, while we agree that a patient’s experience of disability is subjective, when measured with this instrument they do appear to provide additional patient-important information above that provided by standard objective measures.

9. Will it be more prudent to follow up the patients since it is a well-established limb salvage clinic and look at the change in WHODAS (change WHODAS 2.0) with intervention as compared to a cross-sectional study?

We agree. As a follow up to the current cross-sectional study, we recently initiated a prospective cohort study of patients at risk of limb loss (using the same criteria as the current study). This study compares changes in WHODAS 2.0 disability scores with measures of wound healing to determine if advanced wound care in our limb preservation clinic and program improves patients’ estimated disability. In order to outline these next research steps better for readers, we modified the Conclusion section of the manuscript. This now reads:

“Conclusions

In this cross-sectional study, we found that the majority of people at risk of limb loss suffered a substantial burden of disability. Most of them also expressed concern over their wounds and suffered a moderate amount or great deal of wound-related discomfort or distress. The WHODAS 2.0 displayed excellent clinical acceptability among the included patients as well as evidence of criterion validity. Therefore, it may provide additional important patient-centered information among patients at risk of limb loss. Future studies should further validate the instrument in a multicenter fashion. They should also determine how care and interventions (e.g., to enhance wound healing) provided over time to these patients may decrease their burden of disability, pain, and wound-specific concerns.” (page 24)

10. Limb loss is a major concern in terms of economic burden and also is a life-altering event. Hence I strongly encourage the authors to further work on this manuscript and I will be happy to provide further inputs. Looking forward to your reply to these queries and hopefully this helps in improving the work.

Respectfully,

Reviewer

 We thank you for your kind and thoughtful comments on our work. They have improved our manuscript and are greatly appreciated.

Reviewer #2: Ref: PONE-D-21-10113

In the present article entitled “Disability, pain, and wound-specific concerns self-reported by adults at risk of limb loss: a cross-sectional study” Roberts et al., have estimated the burden of disability in patient-reported outcomes (PROs) in patients at risk of limb loss using the World Health Organization Disability Assessment Schedule 2.0 (WHODAS 2.0). While the data collection and WHODAS 2.0 application appear sound to me, here are specific comments.

We thank the Reviewer for their kind and thoughtful review of our manuscript. The comments have improved the study and are greatly appreciated.

1. Title can be made more precise and inclusion of WHODAS 2.0 to it can make it more apt.

We agree. We have therefore added the World Health Organization Disability Assessment Schedule to the title of the manuscript as suggested by the Reviewer. The revised title of the manuscript is:

“Disability, pain, and wound-specific concerns self-reported by adults at risk of limb loss: a cross-sectional study using the World Health Organization Disability Assessment Schedule 2.0”

2. Introduction: secondary objectives which were to determine the clinical acceptability of WHODAS 2.0, and to look for the extent to which various patient and wound characteristics can predict the degree of disability, can be included in the introduction.

We agree with the Reviewer that the primary and secondary objectives of the study should be explicitly stated in the Introduction section of the manuscript. In direct response to the above suggestion by the Reviewer, we therefore included both the study hypotheses and objectives in the final paragraph of the Introduction. This paragraph now reads:

“Given its broad validity, we hypothesized that the WHODAS 2.0 would be acceptable to patients at risk of limb loss when administered in a clinic setting and would allow for quantification of the prevalence and severity of disability among these patients. We also hypothesized that the WHODAS 2.0 would provide unique patient-important information above that provided by characteristics of these patients and their wounds as well as a direct comparison of the degree of disability to those with other medical and surgical conditions. The primary objective of the study was to estimate the prevalence and severity of disability in patients at risk of limb loss using the WHODAS 2.0. Secondary objectives were to determine whether the WHODAS 2.0 was clinically acceptable to these patients and the extent to which patient and wound characteristics, wound-specific concerns, and wound-related discomfort or distress predicted their disability.” (paragraph 2, page 5)

3. Materials and methods: the patients recruited in this study were thought to be at the risk of limb loss. This can be elaborated further so as to mark if any objective criteria were followed to consider the probability of limb loss.

Thank you. We included patients referred to our limb-preservation clinic who were assessed to be at risk of limb loss by an experienced PhD (wound care)-trained specialist wound care nurse and vascular surgeon because there is no well-validated instrument for assessing risk of limb loss among patients with different wound types. The Society for Vascular Surgery (SVS) Wound, Ischemia, foot Infection (SVS WIfI) classification system may have provided an additional objective assessment of the risk of limb loss among those with arterial-insufficient and diabetic wounds. However, the SVS WIfI was not designed for use in patients with other types of hard-to-heal wounds, including those that were included in this study (e.g., chronic venous or postoperative wounds). In order to make this clear for readers, the Participants section of the Methods reads:

“Participants

There is no well-validated instrument for assessing risk of limb loss among patients with different wound types aside from arterial-insufficient and diabetic foot wounds. We therefore included consecutive adults (age >18-years) referred to TOH Limb-Preservation Clinic starting in June 1, 2018 thought to be at risk of limb loss by both the specialty wound care nurse and one of six vascular surgeons with extensive experience in limb-preservation. A vascular surgeon first evaluated all patients before they were seen in clinic. Our goal was to recruit a diverse cohort of patients with hard-to-heal wounds at risk of limb loss.” (paragraph 3, page 7-8)

 We also acknowledge the above as a potential limitation in the limitations section of the Discussion. This limitations section reads:

“Our findings need to be considered in the context of the study’s limitations. First, we included patients referred to our limb-preservation clinic who were assessed to be at risk of limb loss by an experienced PhD (wound care)-trained specialist wound care nurse and vascular surgeon. Although the baseline characteristics of these patients appeared characteristic of patients at risk of limb loss, the Society for Vascular Surgery (SVS) Wound, Ischemia, foot Infection (SVS WIfI) classification system may have provided an additional objective assessment of the risk of limb loss among those with arterial-insufficient and diabetic wounds. However, the SVS WIfI was not designed for use in patients with other types of hard-to-heal wounds (e.g., chronic venous or postoperative wounds). Further, we lacked patient racial and economic data and some of those included in our study did not have arterial pressure measurements required to stratify patients into SVS WIfI stages at their first clinic assessment. Second, while our patients found the WHODAS 2.0 to be clinically acceptable, they did require assistance to input scores into a tablet. Future studies should therefore assess whether findings would be similar when collected via patient-facing data entry. Third, while it may be argued that a number of generic quality of life and disease-specific instruments already exist for assessing PROs in patients at risk of limb loss, the WHODAS 2.0 has been extensively validated, displays broad applicability across those at risk of limb loss, and when combined with measures of wound-specific concerns and discomfort/distress captures all of the domains covered by these other instruments. It also allows for direct comparison between other patient populations and studies of disability. Finally, while the WHODAS 2.0 is widely validated across disease states and demonstrated promising predictive validity in this study, our evaluation did not assess all aspects of validity. Future studies should therefore assess concurrent and convergent validity and perform longitudinal follow-up to determine reliability.” (paragraph 2, page 23-24)

4. Data collection, methods of measurement, and definitions: Limb ischemia is one of the presentations of various systemic diseases like Diabetes, Atherosclerosis, Aortoarteritis, end stage renal disease and a few more. Various other presentations and symptoms of these systemic illnesses can act as confounding factors while patients respond to the WHODAS 2.0 questionnaire. So this aspect of confounding factors needs to be addressed further.

Thank you for this comment. First, we just hope to clarify the context of the Reviewer’s comment. Specifically, at least in our understanding, a confounder is a variable that influences both the likelihood of an exposure or intervention being present, while also being associated with likelihood of outcome. In the current study, we have not evaluated a specific exposure or intervention in relation to our outcome (WHODAS 2.0 disability score), but instead look to understand what variables may be explanatory (or perhaps predictive, although we are not trying to build a clinical prediction model) of disability.

Within this context, we agree that we did not have all possible explanatory or predictive variables available to us. However, we did find that when considering only a patient’s baseline measured covariates (age, sex, comorbidities) we were unable to explain much of the variation in observed disability scores (7%). In contrast, when we looked at objective and self-reported wound specific variables we were able to explain a moderate amount of variation (40%). 

Ultimately we agree that this leaves much variation to be explained, and we now further highlight the limitations in our available measured variables and point to the need for further, prospective and multicenter evaluation.

5. Discussion paragraph can be further delineated in an intelligible manner to improve the flow.

Thank you. We have reviewed the Discussion of the manuscript in detail and modified this after critical input by each of our coauthors to ensure it is delineated in an intelligible manner to improve the flow. We are wondering if you perhaps specifically are referring to the 1st paragraph of the discussion? Please see the updates below. We hope tha this is satisfactory, however, if the Reviewer or Editor have any further suggestions to improve flow, we would be more than happy to make additional changes. The first paragraph of the Discussion now reads:

“In this cross-sectional study of patients at risk of limb loss due to lower limb wounds, we found that almost three-out-of-four suffered from clinically significant disability. These patients had a number of different types of lower limb wounds, including arterial-insufficient, mixed, postoperative, chronic venous, and diabetic wounds. Further, over half of these patients expressed concerns over their wound(s) and suffered a moderate amount or great deal of wound-related discomfort or distress. Importantly, we found that the increasingly well-established clinical acceptability of the WHODAS 2.0 generalized to the older, comorbid patients routinely seen in a limb-preservation clinic. Finally, the WHODAS 2.0 had evidence of providing unique patient-important information as it was only moderately correlated with other patient-reported and objective wound criteria, and was not entirely explained by wound and patient characteristics.” (paragraph 1, page 21)

6. Authors have not discussed any limitations they faced or tackled during this study.

We apologize if the limitations section of the manuscript was not clearly identified. We therefore revised the first sentence of the limitations section of the Discussion such that it reads “Our findings need to be considered in the context of the study’s limitations.” This limitations paragraph now reads:

“Our findings need to be considered in the context of the study’s limitations. First, we included patients referred to our limb-preservation clinic who were assessed to be at risk of limb loss by an experienced PhD (wound care)-trained specialist wound care nurse and vascular surgeon. Although the baseline characteristics of these patients appeared characteristic of patients at risk of limb loss, the Society for Vascular Surgery (SVS) Wound, Ischemia, foot Infection (SVS WIfI) classification system may have provided an additional objective assessment of the risk of limb loss among those with arterial-insufficient and diabetic wounds. However, the SVS WIfI was not designed for use in patients with other types of hard-to-heal wounds (e.g., chronic venous or postoperative wounds). Further, we lacked patient racial, economic, housing, and external social and financial support data, and some of those included in our study did not have arterial pressure measurements required to stratify patients into SVS WIfI stages at their first clinic assessment. Second, while our patients found the WHODAS 2.0 to be clinically acceptable, they did require assistance to input scores into a tablet. Future studies should therefore assess whether findings would be similar when collected via patient-facing data entry. Third, while it may be argued that a number of generic quality of life and disease-specific instruments already exist for assessing PROs in patients at risk of limb loss, the WHODAS 2.0 has been extensively validated, displays broad applicability across those at risk of limb loss, and when combined with measures of wound-specific concerns and discomfort/distress captures all of the domains covered by these other instruments. It also allows for direct comparison between other patient populations and studies of disability. Finally, while the WHODAS 2.0 is widely validated across disease states and demonstrated promising predictive validity in this study, our evaluation did not assess all aspects of validity. Future studies should therefore assess concurrent and convergent validity and perform longitudinal follow-up to determine reliability..” (paragraph 2, page 23-24)

Thank you once again for the reviews. The comments provided by Dr. Singh and the Reviewers have improved our manuscript. We hope that you will find this version suitable for publication in PLoS One, and look forward to your response.

Sincerely,

Derek J. Roberts, MD, PhD and Daniel I. McIsaac, MD, MPH

References

1. von Elm E, Altman DG, Egger M, Pocock SJ, Gotzsche PC, et al. (2008) The Strengthening the Reporting of Observational Studies in Epidemiology (STROBE) statement: guidelines for reporting observational studies. J Clin Epidemiol 61: 344-349.

2. Benchimol EI, Smeeth L, Guttmann A, Harron K, Moher D, et al. (2015) The REporting of studies Conducted using Observational Routinely-collected health Data (RECORD) statement. PLoS Med 12: e1001885.

3. Conijn AP, Jens S, Terwee CB, Breek JC, Koelemay MJ (2015) Assessing the quality of available patient reported outcome measures for intermittent claudication: a systematic review using the COSMIN checklist. Eur J Vasc Endovasc Surg 49: 316-334.

4. Hicks CW, Lum YW (2015) Patient-reported outcome measures in vascular surgery. Semin Vasc Surg 28: 122-133.

5. Alabi O, Roos M, Landry G, Moneta G (2017) Quality-of-life assessment as an outcomes measure in critical limb ischemia. J Vasc Surg 65: 571-578.

6. Olsson M, Jarbrink K, Divakar U, Bajpai R, Upton Z, et al. (2019) The humanistic and economic burden of chronic wounds: A systematic review. Wound Repair Regen 27: 114-125.

7. Khunkaew S, Fernandez R, Sim J (2019) Health-related quality of life among adults living with diabetic foot ulcers: a meta-analysis. Qual Life Res 28: 1413-1427.

8. Phillips P, Lumley E, Duncan R, Aber A, Woods HB, et al. (2018) A systematic review of qualitative research into people's experiences of living with venous leg ulcers. J Adv Nurs 74: 550-563.

9. Steunenberg SL, Raats JW, Te Slaa A, de Vries J, van der Laan L (2016) Quality of Life in Patients Suffering from Critical Limb Ischemia. Ann Vasc Surg 36: 310-319.

10. Bosma J, Vahl A, Wisselink W (2013) Systematic review on health-related quality of life after revascularization and primary amputation in patients with critical limb ischemia. Ann Vasc Surg 27: 1105-1114.

11. Shulman MA, Myles PS, Chan MT, McIlroy DR, Wallace S, et al. (2015) Measurement of disability-free survival after surgery. Anesthesiology 122: 524-536.

12. Abedzadeh-Kalahroudi M, Razi E, Sehat M, Asadi-Lari M (2016) Psychometric properties of the world health organization disability assessment schedule II -12 Item (WHODAS II) in trauma patients. Injury 47: 1104-1108.

13. Lee HH, Shin EK, Shin HI, Yang EJ (2017) Is WHODAS 2.0 Useful for Colorectal Cancer Survivors? Ann Rehabil Med 41: 667-676.

14. Soberg HL, Finset A, Roise O, Bautz-Holter E (2012) The trajectory of physical and mental health from injury to 5 years after multiple trauma: a prospective, longitudinal cohort study. Arch Phys Med Rehabil 93: 765-774.

15. Wolf AC, Tate RL, Lannin NA, Middleton J, Lane-Brown A, et al. (2012) The World Health Organization Disability Assessment Scale, WHODAS II: reliability and validity in the measurement of activity and participation in a spinal cord injury population. J Rehabil Med 44: 747-755.

16. Kutlay S, Kucukdeveci AA, Elhan AH, Oztuna D, Koc N, et al. (2011) Validation of the World Health Organization disability assessment schedule II (WHODAS-II) in patients with osteoarthritis. Rheumatol Int 31: 339-346.

17. Kucukdeveci AA, Kutlay S, Yildizlar D, Oztuna D, Elhan AH, et al. (2013) The reliability and validity of the World Health Organization Disability Assessment Schedule (WHODAS-II) in stroke. Disabil Rehabil 35: 214-220.

18. Schlote A, Richter M, Wunderlich MT, Poppendick U, Moller C, et al. (2009) WHODAS II with people after stroke and their relatives. Disabil Rehabil 31: 855-864.

19. Garin O, Ayuso-Mateos JL, Almansa J, Nieto M, Chatterji S, et al. (2010) Validation of the "World Health Organization Disability Assessment Schedule, WHODAS-2" in patients with chronic diseases. Health Qual Life Outcomes 8: 51.

20. Ustun TB, Chatterji S, Kostanjsek N, Rehm J, Kennedy C, et al. (2010) Developing the World Health Organization Disability Assessment Schedule 2.0. Bull World Health Organ 88: 815-823.

---

## [Decision Letter · Decision Letter 1]

2 Jun 2021

Disability, pain, and wound-specific concerns self-reported by adults at risk of limb loss: A cross-sectional study using the World Health Organization Disability Assessment Schedule 2.0

PONE-D-21-10113R1

Dear Dr. Roberts,

We’re pleased to inform you that your manuscript has been judged scientifically suitable for publication and will be formally accepted for publication once it meets all outstanding technical requirements.

Kind regards,

Kanhaiya Singh, Ph.D

Academic Editor

PLOS ONE

Additional Editor Comments (optional):

Reviewers' comments:

Reviewer's Responses to Questions

**Comments to the Author**

1. If the authors have adequately addressed your comments raised in a previous round of review and you feel that this manuscript is now acceptable for publication, you may indicate that here to bypass the “Comments to the Author” section, enter your conflict of interest statement in the “Confidential to Editor” section, and submit your "Accept" recommendation.

Reviewer #1: All comments have been addressed

Reviewer #2: All comments have been addressed

2. Is the manuscript technically sound, and do the data support the conclusions?

Reviewer #1: Yes

Reviewer #2: Yes

3. Has the statistical analysis been performed appropriately and rigorously? 

Reviewer #1: Yes

Reviewer #2: N/A

4. Have the authors made all data underlying the findings in their manuscript fully available?

Reviewer #1: Yes

Reviewer #2: Yes

5. Is the manuscript presented in an intelligible fashion and written in standard English?

Reviewer #1: Yes

Reviewer #2: Yes

6. Review Comments to the Author

Reviewer #1: (No Response)

Reviewer #2: (No Response)

7. PLOS authors have the option to publish the peer review history of their article (what does this mean?). If published, this will include your full peer review and any attached files.

Reviewer #1: No

Reviewer #2: No

---

## [Editor Report · Acceptance letter]

7 Jun 2021

PONE-D-21-10113R1 

Disability, pain, and wound-specific concerns self-reported by adults at risk of limb loss: A cross-sectional study using the World Health Organization Disability Assessment Schedule 2.0 

Dear Dr. Roberts:

I'm pleased to inform you that your manuscript has been deemed suitable for publication in PLOS ONE. Congratulations! Your manuscript is now with our production department. 

Kind regards, 

on behalf of

Dr. Kanhaiya Singh 

Academic Editor

PLOS ONE